# Avoided metallicity in a hole-doped Mott insulator on a triangular lattice

Chi Ming Yim [1,2,8] ✉, Gesa-R. Siemann [1,8], Srdjan Stavrić [3,4,8], Seunghyun Khim [5], Izidor Benedičič [1], Philip A. E. Murgatroyd[1], Tommaso Antonelli[1], Matthew D. Watson [6], Andrew P. Mackenzie [1,5], Silvia Picozzi[3] ✉, Phil D. C. King [1] ✉ & Peter Wahl [1,7] ✉

Doping of a Mott insulator gives rise to a wide variety of exotic emergent states, from high-temperature superconductivity to charge, spin, and orbital orders. The physics underpinning their evolution is, however, poorly understood. A major challenge is the chemical complexity associated with traditional routes to doping. Here, we study the Mott insulating $CrO_2$ layer of the delafossite $PdCrO_2$, where an intrinsic polar catastrophe provides a clean route to doping of the surface. From scanning tunnelling microscopy and angle-resolved photoemission, we find that the surface stays insulating accompanied by a short-range ordered state. From density functional theory, we demonstrate how the formation of charge disproportionation results in an insulating ground state of the surface that is disparate from the hidden Mott insulator in the bulk. We demonstrate that voltage pulses induce local modifications to this state which relax over tens of minutes, pointing to a glassy nature of the charge order.

The Mott–Hubbard Hamiltonian is one of the simplest models to describe correlated electron physics, capturing phenomena ranging from antiferromagnetic order in a Mott insulator to potentially explaining the high-temperature superconductivity in cuprates. It yields particularly exciting predictions for systems on triangular lattices, including, for a single-orbital Hubbard model, the famed resonating valence bond state[1], and the formation of quantum spin liquids and complex magnetic orders[2]. In multi-orbital systems, the situation is even richer: for example, for transition metal atoms with a partially filled $t_{2g}$ manifold, the interplay of spin–orbit coupling with correlations can result in topologically non-trivial fractional Chern states[3,4], while charge carrier doping can lead to exotic superconducting states[5]. This large variety of unconventional ground states motivates the study of triangular-lattice Mott systems experimentally, and in particular probing the evolution of their ground states with doping.

Here, we establish the surface of $PdCrO_2$ as a model system in which to investigate the competing ground states of a doped triangular-lattice Mott system. Its bulk crystal structure (Fig. 1a) consists of stacked triangular-lattice $Pd^{1+}$ and $(CrO_2)^{1-}$ layers. The former is in a $4d^9$ charge state, forming highly conductive metallic layers[6–9]. In contrast, the latter host $Cr^{3+}$ ions, with a $3d^3$ electron configuration which half-fills the $t_{2g}$ manifold. These layers are Mott insulating, and develop an $S = 3/2$ antiferromagnetic (AF) order below a Néel temperature of $T_N = 37.5\,K$[8,10–12]. This naturally occurring heterostructure of metallic and Mott-insulating layers makes this "hidden" Mott state[12] an ideal candidate for detailed spectroscopic study[8].

[1]SUPA, School of Physics and Astronomy, University of St Andrews, North Haugh, St Andrews, Fife KY16 9SS, UK. [2]Tsung Dao Lee Institute and School of Physics and Astronomy, Shanghai Jiao Tong University, 201210 Shanghai, China. [3]Consiglio Nazionale delle Ricerche (CNR-SPIN), Unitá di Ricerca presso Terzi c/o Universitá "G. D'Annunzio", 66100 Chieti, Italy. [4]Vinča Institute of Nuclear Sciences -National Institute of the Republic of Serbia, University of Belgrade, P. O. Box 522, RS-11001 Belgrade, Serbia. [5]Max Planck Institute for Chemical Physics of Solids, Nöthnitzer Straße 40, 01187 Dresden, Germany. [6]Diamond Light Source, Harwell Science and Innovation Campus, Didcot OX11 ODE, UK. [7]Physikalisches Institut, Universität Bonn, Nussallee 12, 53115 Bonn, Germany. [8]These authors contributed equally: Chi Ming Yim, Gesa-R. Siemann, Srdjan Stavrić. ✉e-mail: c.m.yim@sjtu.edu.cn; silvia.picozzi@spin.cnr.it; pdk6@st-andrews.ac.uk; wahl@st-andrews.ac.uk

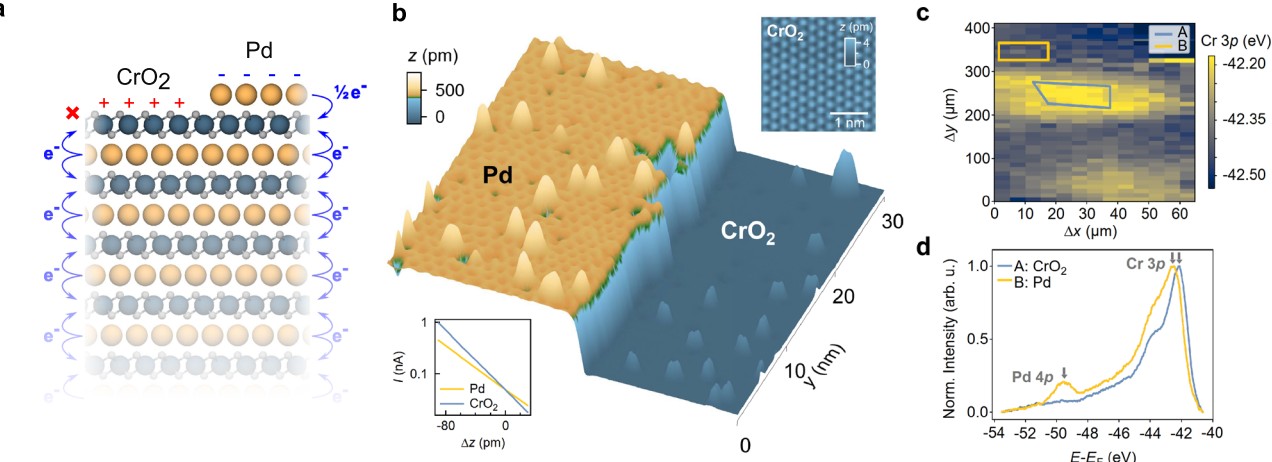

**Fig. 1 | Surface terminations of the delafossite PdCrO2. a** Crystal structure of PdCrO2 showing two possible surface terminations: Pd and CrO2. Electrons are transferred from Pd layers to CrO2 layers in the bulk. For the surface CrO2 layer, only one Pd layer donates electrons to it, resulting in effective hole-doping of the surface layer. Similarly, a surface Pd layer donates electrons to only one CrO2 layer underneath it, leading to effective electron doping. **b** 3D rendered STM topograph of a freshly cleaved surface of a PdCrO2 single crystal ($V = 0.8$ V, $I = 50$ pA; image size $(30$ nm$)^2$). The imaged region comprises a Pd-terminated terrace on the left, a CrO2-terminated terrace on the right, and a step-edge separating the two. Inset (bottom left), $I(\Delta z)$ curves recorded from the two terraces, showing that the CrO2 terrace has a larger apparent work function (see also Supplementary Note 1, Supplementary Fig. 1). Inset (top-right), atomically resolved STM image of the CrO2 terrace showing an unreconstructed $(1 \times 1)$ unit cell ($V = 300$ mV, $I = 50$ pA; image size $(3$ nm$)^2$), scale bar: 1 nm. **c** Spatially resolved soft X-ray photoemission spectroscopy (XPS) map ($h\nu = 110$ eV) taken over the energy range of the Cr $3p$ and Pd $4p$ core levels (arrows in **d**) showing shifts in the binding energy of the Cr $3p$ core level when moving across the sample. **d** Core-level data extracted from the regions in **c** corresponding to the two distinct surface terminations CrO2 (A: blue region in **c**) and Pd (B: yellow region in **c**).

The alternating $+1/-1$ nominal valences of the Pd and CrO2 layers reflect a net charge transfer of one electron from Pd to the neighbouring CrO2 layers in the bulk (cf. Fig. 1a). The loss of such charge transfer processes at the surface, however, is expected−in a simple ionic picture−to result in an approximately 0.5 holes/Cr (0.5 electrons/Pd) self-doping of the CrO2 (Pd) terminated surface. Such electronic reconstructions−akin to models for the SrTiO3/LaAlO3 interfacial electron gas[13]− have already been observed for the CoO2-based delafossites, resulting in the stabilization of Rashba-coupled and ferromagnetic metallic surface states on the CoO2 and Pd terminated surfaces, respectively[14−18]. For the CrO2-terminated surface of PdCrO2, the same mechanism would naively be expected to mediate a substantial doping of 0.5 holes/Cr of the Mott insulating state found in the bulk and drive an insulator-metal transition. In contrast, we show here that the doping triggers a charge-disproportionation in the surface layer, inducing a structural corrugation distinct from the bulk, and driving the surface layer insulating. We further demonstrate that the charge disproportionated state exhibits short-range order and glassy dynamics, evidencing a high degree of degeneracy of the ground state.

## Results

### Determination of surface termination

Due to the stronger bonding between Cr and O within the CrO2 octahedra as compared to the bond between oxygen and palladium atoms, PdCrO2 is expected to cleave between Pd and O. This results in two surface terminations that leave the in-plane order intact−a Pd and a CrO2 terminated one (Fig. 1a). Consistent with this simple picture, we show in Fig. 1b a topographic scanning tunnelling microscopy (STM, see the "Methods" section) image of the surface of a freshly cleaved sample. The region comprises two flat terraces separated by a step edge. The step height is only ∼480 pm, less than the 600 pm expected for adjacent terraces of identical surface termination. This, as well as the different corrugations of the surfaces, suggests that the step edge here is between surfaces of distinct atomic characters. Similarly, the presence of two different surface terminations is also evidenced by our spatially resolved photoemission measurements, performed using a

light spot of ∼4 μm in lateral size (see the "Methods" section). The small spot size and typical terrace sizes seen in STM images (see Supplementary Fig. 3), as well as our experience with the surface terminations of other delafossites[15,16] suggest that in most surface regions, there is no significant inhomogeneity within the probed area. We show in Fig. 1c the spatial variations of the binding energy of the Cr $3p$ core level, imaged over a region of $(400 \times 65)$ μm$^2$ of the cleaved surface. Characteristic core-level shifts of ∼250 meV are observed, varying between patches of the sample with lateral sizes of ≈50−100 μm. The extracted core level spectra integrated over the regions shown in Fig. 1c are shown in Fig. 1d, spanning both the Cr $3p$ and Pd $4p$ core levels. The spectrum which exhibits the higher binding energy of the Cr $3p$ core level (region B, orange line) also exhibits a much more substantial Pd $4p$ peak, which is almost completely absent in the spectrum from region A (blue line), similar to termination-dependent core-level spectra from the related material AgCrSe2[19]. Both, the core-level shift of the Cr peak as well as the variation of the intensity of the Pd peak allow us to assign region A as CrO2 terminated and B as Pd terminated. The complete absence of the Pd $4p$ peak on the CrO2-terminated areas is likely due to a combination of the lower photoemission cross-section of this peak and the finite probing depth in our measurements. In our atomically-resolved STM data, $I(\Delta z)$ curves (bottom-left inset of Fig. 1b) likewise show different apparent barrier heights for different surface terminations. From the STM data, we assign the surface regions with lower apparent barrier height as Pd-terminated surface regions (region B), and those exhibiting a larger apparent barrier height as CrO2-terminated areas (region A). These assignments allow us to directly link spectroscopic data between the STM and ARPES data of the two terminations. We will in the following focus on region B, the CrO2-terminated surface.

### Near-Fermi level electronic structure of the CrO2-terminated surface

The reduced binding energy of the Cr $3p$ core-level for the CrO2-terminated surface already provides spectroscopic evidence for the expected hole doping of the surface CrO2 layer introduced above. To probe the influence of this on the low-energy electronic structure, we

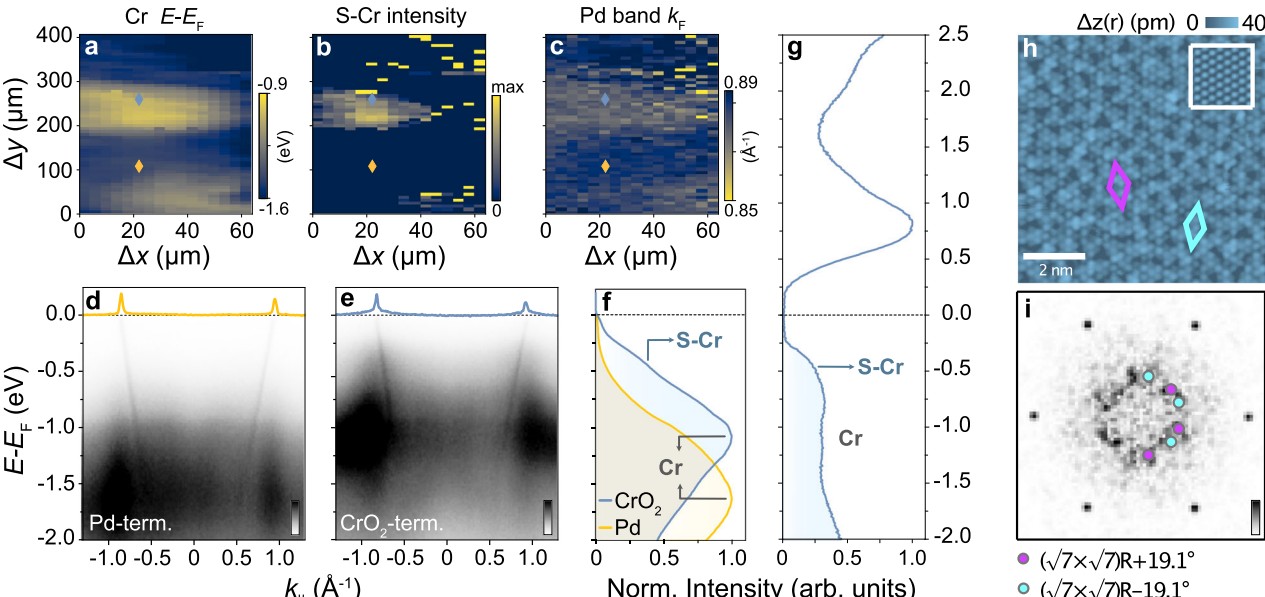

**Fig. 2 | Termination-dependent electronic structures. a–f** Spatially resolved angle-resolved photoemission spectroscopy (ARPES) data of the valence band electronic structure, measured using a photon energy of $h\nu = 110$ eV and linear horizontal (LH) polarized light. Spatially resolved mapping data revealing changes in **a** the binding energy ($E-E_F$) of the Cr-derived peak in the valence band (labelled Cr in **f**), **b** the spectral weight of the Cr-shoulder (S-Cr), and **c** the Fermi wave vector $k_F$ of the steep Pd-derived states, measured across the PdCrO₂ surface over the same region as that in Fig. 1c. **d, e** ARPES dispersions of a **d** Pd-terminated (yellow marker in the spatial maps) and **e** CrO₂-terminated (blue marker in the spatial maps) surface region and **f** the $k$-integrated spectra integrated over the full $k$-range of the dispersions shown in **d** and **e**. **g** Differential conductance spectrum $g(V)$ of the CrO₂ terminated surface measured with STM ($V_s = 1.5$ V, $I_s = 50$ pA; $V_m = 10$ mV), showing a shoulder at the same energy as the S-Cr feature in **f**, as well as the upper edge of the surface gap at +0.25 eV. **h** Atomically resolved STM topograph of the CrO₂ terminated surface recorded with a bias voltage in the gap near $E_F$ in **g** ($V = 15$ mV, $I = 150$ fA; image size (8 nm)²). Inset, the image at a bias voltage within the peak above $E_F$ in **g** ($V = 350$ mV, $I = 150$ pA; image size (2 nm)²). Coloured rhombi indicate characteristic building blocks of the short-range order with unit cells of $(\sqrt{7} \times \sqrt{7})R \pm 19.1°$ formed locally within the CrO₂ layer. **i** Corresponding Fourier transform of (**h**). Coloured dots mark the FFT peaks of the $(\sqrt{7} \times \sqrt{7})R \pm 19.1°$ order.

show in Fig. 2a–f spatial- and angle-resolved photoemission measurements performed over the same spatial region as the core-level mapping in Fig. 1c. Figure 2d and e show the measured electronic dispersion of a Pd and CrO₂ terminated region, respectively, as identified from our core level mapping in Fig. 1c (see yellow and blue markers in Fig. 2a–c for the measurement positions). In both regions, we find highly dispersive bands derived from the bulk-like Pd states[6–8], which cross the Fermi level ($E_F$) at a Fermi wave vector $k_F \sim \pm 0.87$ Å⁻¹. From the fitting of momentum distribution curves (MDCs) at $E_F$, we find a small but non-negligible decrease of $k_F$ for the CrO₂- versus the Pd-terminated surface regions (Fig. 2c, Supplementary Fig. 4), indicating a weak hole doping of these Pd-derived states for the CrO₂-terminated surface.

In addition to these metallic states, we observe rather diffuse spectral weight gapped away from $E_F$, which varies much more significantly between the two surface terminations. We attribute this spectral weight to states of Cr-derived character. For the Pd-terminated regions, this diffuse weight is peaked at a binding energy of ≈1.55 eV (evident in the $k$-integrated spectrum in Fig. 2f), consistent with the location of the lower Hubbard band, which derives from the bulk CrO₂ Mott insulating layer[8]. For the CrO₂-terminated surface, however, we find that this component shifts towards lower binding energies by around 500 meV, while a pronounced shoulder (S-Cr in Fig. 2f) develops closer to the Fermi level at $E-E_F \approx 0.5$ eV. All of these spectral features—the shift to lower binding energy of the dominant Cr-derived density of states (DOS) peak (Fig. 2a), development of spectral weight of the lower-energy shoulder (S-Cr, Fig. 2b), and decrease in $k_F$ of the Pd-derived band (Fig. 2c)—show a spatial dependence that is closely correlated with our identification of CrO₂-derived regions from our core-level spatial mapping (Fig. 1c), indicating that they are characteristic signatures of the electronic structure of the CrO₂-terminated surface.

A direct comparison to our tunnelling spectra measured on the CrO₂-terminated surface, shown in Fig. 2g and Supplementary Fig. 5, reveals similar features. The differential conductance spectrum $g(V)$ shows a clear gap-like structure, with gap edges at ±0.25 eV, vanishingly small DOS inside the gap, and onset of tunnelling into the occupied states and a broad maximum that is in good agreement with the S-Cr feature observed in ARPES. Taken together, our surface-sensitive spectroscopic measurements thus demonstrate clear modifications in the electronic structure of the CrO₂-terminated surface from that of the Mott-insulating CrO₂ layers of the bulk. Nonetheless, despite the large doping away from half-filling, which is inherent to this polar surface, our measurements show that it remains insulating, defying the simple expectation for such a heavily doped Mott insulator. This suggests that a different form of correlated insulating state is obtained here.

While the differential conductance measured by STM inside the surface gap becomes almost zero, tunnelling through the surface layer to the Pd layer underneath remains possible, although it requires extremely small tunnelling currents (below 500 fA) to achieve stable tunnelling. Excitingly, this enables detailed probing of the correlated states of the surface layer, which would otherwise be extremely challenging for such an insulating layer. Atomically resolved STM measurements performed at a bias voltage of 0.3 V, outside the gap, show an unreconstructed, perfect triangular lattice with a (1 × 1) unit cell (inset in Fig. 1b). We find a notable change in the appearance of this surface topography, however, when imaging the surface at energies inside the gap (Fig. 2h). An additional contrast to that of the regular Cr lattice is evident, which appears rather disordered. The point defects, clearly visible in the topography in Fig. 1b, become virtually invisible when imaged at low energies (see Supplementary Note 5, Supplementary Fig. 6). From the real-space image and its Fourier transform (Fig. 2i), we can identify a characteristic length scale, indicating the

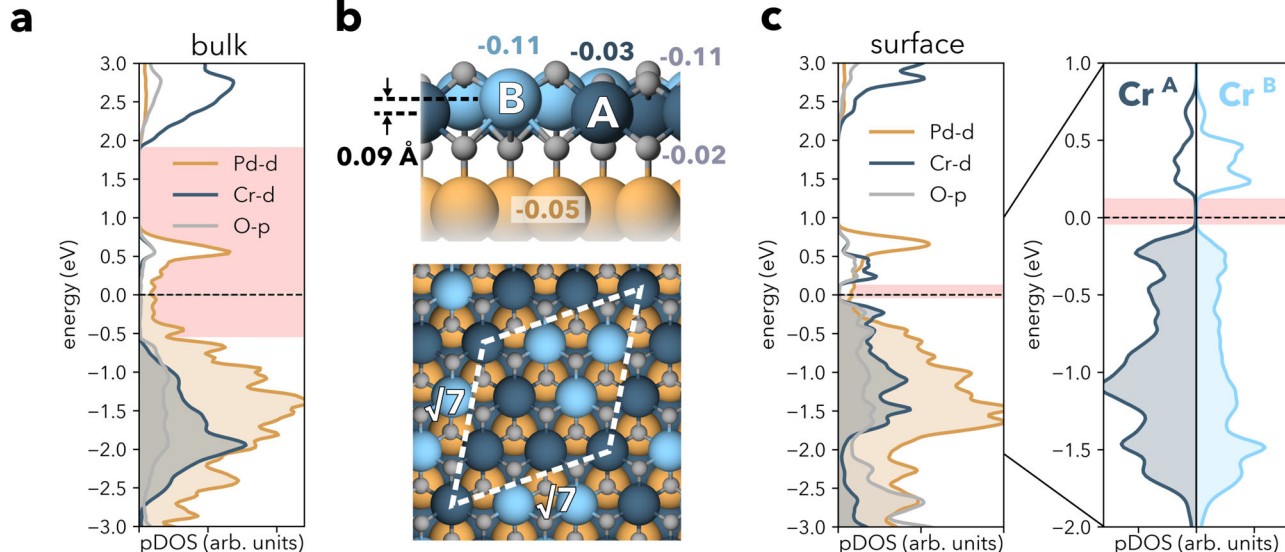

**Fig. 3 | Modelling of the CrO$_2$ surface. a** Projected density of states (PDOS) of bulk PdCrO$_2$ modelled with $\sqrt{7} \times \sqrt{7}$ unit cell and a collinear configuration of 4 spin-up and 3 spin-down Cr atoms (see the "Methods" section), exhibiting the metallic nature of Pd states and the hidden Mott insulating nature of CrO$_2$ layers (the gap is emphasized with red stripe). **b** CrO$_2$ surface modelled with the same unit cell and relaxed, showing a vertical displacement of Cr atoms of up to ~0.1 Å and resulting in two distinct types of Cr atoms, Cr$^A$ and Cr$^B$. The numbers near spheres represent the difference in electronic charge of the atoms at the surface from those in the bulk obtained from a Bader analysis (negative values reflect hole doping). **c** PDOS of the CrO$_2$ surface. The states at $E_F$ are derived from the subsurface Pd layer, whereas the Cr $t_{2g}$ states are gapped by ~0.2 eV, with Cr$^B$ more strongly hole-doped than Cr$^A$. The inset to the right shows the PDOS for Cr$^A$ and Cr$^B$ for a narrower energy range around the Fermi energy.

formation of a short-range ordered state. From the dominant features in the Fourier transform, we identify the basic building block of this short-range order as a $(\sqrt{7} \times \sqrt{7})R \pm 19.1°$ unit.

## Formation of a charge disproportionated insulator

To understand why the surface layer remains insulating, despite the significant charge transfer and the origins of its peculiar local order, we have modelled a PdCrO$_2$ slab and performed spin-polarized density functional theory (DFT) calculations. We use a DFT + $U$ scheme to include the effect of correlations—a calculation scheme that we find well reproduces the hidden Mott gap of the bulk electronic structure, Fig. 3a (see Supplementary Note 6 for details).

We have modelled the CrO$_2$ surface by a slab containing three Pd and four CrO$_2$ layers in a $\sqrt{3} \times \sqrt{3}$ unit cell enclosed by a vacuum. Calculations using such a slab, including relaxation, stabilize a ferromagnetic ground state in the surface CrO$_2$ layer that is metallic—as expected for a heavily-doped Mott insulator, but clearly inconsistent with our spectroscopic data. To address this discrepancy, we introduce the peculiar $\sqrt{7} \times \sqrt{7}$ order observed in our STM measurements into the calculations by creating a unit cell containing seven Cr atoms per layer. Relaxing the atomic positions in this unit cell results in a structural modification of the surface layer, which leads to two distinct types of Cr atoms, Cr$^A$ and Cr$^B$, which are distributed with a 4:3 ratio and arranged in a pattern as depicted in Fig. 3b.

Cr atoms of both types relax outwards at the surface but by different amounts. The relaxation results in an off-centring of the Cr atoms within the surface CrO$_6$ octahedra, by 0.06 and 0.14 Å for Cr$^A$ and Cr$^B$ atoms, respectively, with the larger out-of-plane relaxation of Cr$^B$ accompanied by a larger distortion of the octahedra. The difference of ~0.1 Å in the height of the two Cr species provides a natural explanation for the contrast in the STM images evident in Fig. 2h (compare also Supplementary Fig. 10 for simulated STM images).

Most strikingly, this structural distortion renders the surface insulating, stabilizing a full gap of ~0.2 eV within the Cr $t_{2g}$ manifold, as depicted in Fig. 3c. In agreement with our STM and ARPES measurements, this gap is substantially smaller when compared to the bulk gap which separates the $t_{2g}$ and $e_g$ manifolds. This is in stark contrast with calculations for an unreconstructed surface, where DFT calculations yield a metallic state (see Supplementary Note 8).

The key question is, therefore, why the CrO$_2$ surface avoids such a metallic state and how the insulating state results in the reconstructed surface layer. In a purely ionic picture (Fig. 1a), simple electron counting would suggest that the 0.5 hole/Cr doping of the surface layer could be accommodated via a charge ordering of the Cr sites, with equal numbers of $d^2$ and $d^3$ occupancies to give an average $d^{2.5}$ configuration. In line with this picture, we find that the charges in the surface CrO$_2$ layer redistribute laterally, resulting in Cr$^A$ atoms containing more electrons in their $t_{2g}$ manifold than Cr$^B$ atoms (upper panel in Fig. 3b). This is reflected by a higher magnetic moment of the Cr$^A$ sites of 2.83 $\mu_B$ (close to the 2.89 $\mu_B$ of the bulk) as compared to 2.49 $\mu_B$ of the Cr$^B$ atoms. A Bader charge analysis provides additional support for such a picture, with Cr$^A$ (Cr$^B$) atoms having 0.03 (0.11) fewer electrons than the Cr atoms in the bulk (Fig. 3b). The Pd atoms in the layer underneath remain non-magnetic and with the same charge throughout. The charge disproportionation that occurs in the surface CrO$_2$ layer only happens once the on-site Coulomb interaction is added to the Cr 3$d$ electrons through the Hubbard-like $U$ term, suggesting a critical role of electron correlations. We also note that, because the wave functions are delocalized also over the oxygen atoms, an additional contribution to the charge order might also come from the oxygen ligands.

Indeed, our analysis indicates that the additional surface charge is not purely accommodated by the surface Cr atoms. The topmost oxygen atoms, sitting above the Cr layer, have fewer electrons than those in the bulk (Fig. 3b). Furthermore, the doping is not confined exclusively to the surface CrO$_2$ layer, and we find a slight hole doping (0.05 holes/Pd) for the subsurface Pd layer, as evidenced by a shift of its Pd band crossing $E_F$ by ~0.1 eV towards lower binding energies compared to the Pd bands in deeper layers (see Supplementary Note 10, Supplementary Fig. 13a, b). In consequence, there are less than 0.5 holes/Cr left to be distributed among the Cr atoms and slightly more than half of the Cr atoms (Cr$^A$) stay in a bulk-like $d^3$ configuration

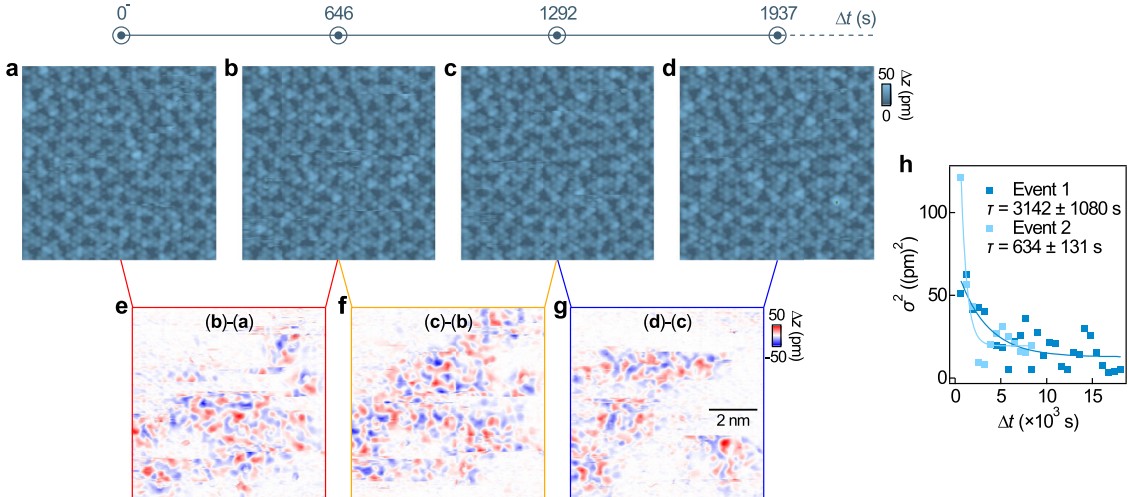

**Fig. 4 | Dynamics of the glassy state in the surface CrO₂ layer. a–d** Atomically resolved images of the CrO₂-terminated surface taken (**a**) before and (**b**)–(**d**) after a 70 mV voltage pulse with a duration of 2 s were applied in the centre of the image. ($V = 5$ mV, $I = 150$ fA, image size: $(8$ nm$)^2$). The time $\Delta t$ after the voltage pulse at which the images were obtained is indicated. **e**–**g** Difference of consecutive images shown in (**a**)–(**d**) reveal the dynamics of the charge ordered state following the voltage pulse. **h** Scatter plot of the variance of the height distribution in the difference images as a function of $\Delta t$. An exponential fit to the data recorded from two individual experiments reveals relaxation time constants ($\tau$) of ~3140 ± 1080 s (dark blue) and ~634 ± 131 s (light blue), respectively.

(as inferred from their magnetic moment), while the remainder of the Cr atoms (Cr$^B$) end up in a configuration as close as possible to $d^2$. This charge disproportionation lowers the energy when compared to a metallic state, where charges are delocalized and homogeneously distributed among the atoms of the same species. The charge disproportionation further results in the gapping of the electronic states. The Mott gap becomes replaced by a charge gap driven by the localization of charge carriers and Coulomb repulsion. Other mechanisms, such as the formation of a charge density wave, would not result in complete gapping of the Fermi surface.

The charge disproportionation and the accompanying structural modification are thus instrumental in avoiding metallicity and maintaining the insulating nature of the surface CrO₂ layer. Also, we find that details of the magnetic order on the Cr atoms only affect the size of the gap, but not its presence, hence suggesting that charge disproportionation is the primary driver for the insulating ground state (see Supplementary Note 8, Supplementary Fig. 12). In this respect, we note that, for the bulk, the opening of the Mott gap does not require magnetism to be included in calculations once correlation effects are properly accounted for[12,20]. The robustness of the gap in the surface layer to details of the magnetic order suggests that the situation there mirrors that in the bulk, and treatments with methods that account more realistically for correlation effects may yield an insulating state without magnetic order.

### Glassy dynamics

While our calculations and experiments demonstrate that the doped Mott layer at the surface of PdCrO₂ forms a charge-disproportionated insulator, both data and experiments also point towards a competition of ground states with similar periodicity. In the experiments, this is evidenced by the short-ranged nature of the order, suggesting that it is frustrated and only presents a shallow energy minimum, while in calculations, a number of different magnetic ground states are close by in energy. In STM images, the structural modification can only be imaged at extremely small tunnelling currents $I \sim 150$ fA (Fig. 2h), whereas using higher currents triggers changes in the order and STM images appear fuzzy (see Supplementary Note 10, Supplementary Fig. 14, also Supplementary Fig. 6), suggesting that the ground state is susceptible to small perturbations.

Moreover, we find that changes in the surface order can be triggered through deliberate voltage pulses, disrupting the short-range order as seen through changes in the topographic appearance imaged at very low tunnelling currents. The system slowly relaxes towards a new equilibrium, enabling us to study the resulting temporal dynamics of the system. Figure 4 shows such a measurement series. In Fig. 4a, we show the appearance of the short-range order of the surface layer before applying a voltage pulse in the centre of the image. Figure 4b–d shows the time evolution after the voltage pulse has been applied. The images have all been recorded in the same spatial location, with changes in the image contrast reflecting a reconfiguration of the short-range order. Figure 4e–g shows the difference images of the topographic measurements recorded at consecutive time points to highlight the changes.

An analysis of the temporal behaviour from difference images, such as the ones shown in Fig. 4e–g, is shown in Fig. 4h. We find that the relaxation dynamics follow an exponential behaviour with a time constant on the order of tens of minutes. This indicates a slow characteristic timescale over which the system settles following excitation. Moreover, our data indicates that the system relaxes to a different state than the one before the trigger (see Supplementary Note 11, Supplementary Fig. 15). This suggests that the ground state here is only metastable and is characterized by glassy behaviour.

### Discussion

Our study reveals a surprising resilience of the insulating state of the CrO₂ surface layer in the hidden Mott insulator PdCrO₂ to pronounced charge carrier doping away from half-filling. We show how, in the CrO₂ surface layer, a glassy state is formed, developing from a charge-disproportionated insulating ground state. The gap of this state could easily be taken for that of a Mott insulator at half-filling[21], which, however, is not expected here due to the significant charge transfer occurring near the surface. The formation of a charge disproportionation is unexpected here because it occurs in a metal, supporting a description of the system as a heterostructure combining layers with disparate ground states that are largely—although not completely[8]—decoupled, with the screening that would counteract charge disproportionation sufficiently localized in the Pd layer here.

The short-range nature of the charge order and its glassy dynamics suggest a high level of frustration with the relevant interactions and a ground state that is close in energy to multiple nearly degenerate metastable states. The occurrence of charge-ordered states has been widely discussed for the Hubbard model on a square lattice for the cuprates, where a spin-charge separation has been predicted[22] and observed[23] for characteristic dopings, and which exhibit charge-density-wave order in the underdoped material[24,25]. Here, the charge order occurs at significantly higher doping and results in a fully insulating state. This motivates further studies of the $CrO_2$ surface layer, in particular if the charge doping in the surface layer can be further modified by surface doping methods. An interesting question is whether a metallic state can then be recovered. This would provide an exciting platform to study the phase diagram of the resulting correlated states when doping a Mott insulator on a triangular lattice. The metallicity of the sub-surface Pd layer here further ensures that the multi-orbital Mott–Hubbard physics remains accessible for state-of-the-art surface spectroscopies. Such a platform would enable searching for the formation of long-range ordered states, exploring the metal-insulator transition, the interplay with the putative ferromagnetic metal state, as well as searching for the possible emergence of unconventional superconductivity.

## Methods

### Crystal growth
Single-crystal samples of $PdCrO_2$ were grown by the NaCl-flux method as reported in ref. 10. First, polycrystalline $PdCrO_2$ powder was prepared from the following reaction at 960 °C for four days in an evacuated quartz ampoule:

$$2\,LiCrO_2 + Pd + PdCl_2 \rightarrow 2\,PdCrO_2 + 2\,LiCl\,. \tag{1}$$

The obtained powder was washed with water and aqua regia to remove LiCl. The polycrystalline $PdCrO_2$ and NaCl were mixed in the molar ratio of 1:10. Then, the mixture in a sealed quartz tube was heated at 900 °C and slowly cooled down to 750 °C. $PdCrO_2$ single crystals were harvested after dissolving the NaCl flux with water.

### Scanning tunnelling microscopy/spectroscopy
The STM experiments were performed using a home-built low-temperature STM that operates at a base temperature of 1.8 K[26]. Pt/Ir tips were used and conditioned by field emission with a gold target. The bias voltage $V$ is applied to the sample with the tip at virtual ground. Differential conductance ($g(\mathbf{r}, V)$) maps and single point $g(V)$ spectra were recorded using a standard lock-in technique, with the frequency of the bias modulation set at 413 Hz. To obtain a clean surface for STM measurements, $PdCrO_2$ samples were cleaved in situ at temperatures below -20 K in cryogenic vacuum. Measurements were performed at a sample temperature of 4.2 K unless stated otherwise. To be able to image within the surface gap (Fig. 2h), extremely small currents down to about 100 fA have to be used.

### μ-ARPES
The single crystal samples were cleaved in situ at a base pressure lower than $10^{-10}$ mbar. The photoemission measurements were performed using the nano-ARPES branch of the I05 endstation at Diamond Light Source, UK. The light was focussed onto the sample using a reflective capillary optic, which provides a spot size of ~4 μm. The sample was mounted on a five-axis manipulator with piezoelectrically-driven axes used for spatial mapping of the sample. The emitted photoelectrons were detected using a Scienta Omicron DA30 analyser. All measurements were taken at a base temperature of ~ 40 K, using $p$-polarized light with photon energies of $h\nu = 110$ eV for the valence band spectra and $h\nu = 150$ eV for the core level mapping.

### Computational details
Density functional theory (DFT) calculations were performed using the vasp package[27]. Bulk $PdCrO_2$ was modelled assuming the $R\bar{3}m$ space group and using the experimental lattice structure obtained from neutron diffraction at room temperature ($a = 2.9280$ Å, $c = 18.1217$ Å, $z = 0.11057$) with atoms placed at Pd 3a (0, 0, 0), Cr 3b (0, 0, 0.5), and O 6c (0, 0, $z$) positions[11]. We used the unit cell containing six $PdCrO_2$ layers and having the $\sqrt{3} \times \sqrt{3}$ in-layer periodicity (3 Cr atoms per layer) to allow for the modelling of different antiferromagnetic spin arrangements on a triangular Cr lattice.

As discussed in the literature, the correct description of the Mott-insulating electronic structure of $CrO_2$ layers in bulk $PdCrO_2$ requires either the use of a combination of DFT with Dynamic Mean Field Theory (DFT + DMFT) or the extension of the DFT treatment with Hubbard $U$ (and Hund $J$) terms added to the Cr 3$d$ orbitals (DFT + $U$ method)[12,28,29]. We turned to the second option, which is computationally more affordable and allows treating the surfaces. We used the Perdew–Burke–Ernzerhof (PBE) exchange-correlation functional[30] and the rotationally invariant DFT + $U$ approach of Dudarev[31] with parameters $U = 4$ eV and $J = 0.9$ eV which showed the best agreement with experiments and DMFT calculations[8,12]. The details justifying this particular choice of $U$ and $J$ values are discussed in Supplementary Note 6. The plane wave cutoff was set to 350 eV, and the self-consistent field calculations were performed on a (8, 8, 1) mesh of $k$-points.

The $CrO_2$ surface was modelled using a symmetric slab containing three Pd and four $CrO_2$ layers, terminated by $CrO_2$ layers on both ends, in a $\sqrt{7} \times \sqrt{7}$ unit cell. This choice of the unit cell is motivated by our STM measurements, which showed a peculiar $\sqrt{7} \times \sqrt{7}$ periodicity of the surface. Because the structural optimization of large systems in noncollinear approach is burdensome and often leads to numerical instabilities, we turned to computationally more affordable spin-polarized DFT calculations, neglecting spin-orbit coupling. We checked that the bulk hidden Mott gap is reproduced in these calculations (Supplementary Fig. 9). The magnetic moments on Cr atoms in a spin-polarized approach are distributed as follows: the first layer (at the bottom) contains four $+3\mu_B$ and three $-3\mu_B$ magnetic moments, the second layer three $+3\mu_B$ and four $-3\mu_B$, and the third layer has again four $+3\mu_B$ and three $-3\mu_B$. Within a $\sqrt{7} \times \sqrt{7}$ unit cell, such a spin configuration is the closest possible approximation to the AF ordering. Concerning the surface $CrO_2$ layer (the top layer), we tried a few different initial spin configurations and fully relaxed the atomic positions until the forces on all the atoms from the surface $CrO_2$ and subsurface Pd layer dropped below 0.005 eV/Å. The atoms from the other layers are kept fixed in their positions. While the gap in the surface layer is observed for different spin configurations (Supplementary Fig. 12), Supplementary Fig. 11 shows that it only opens once the structural modification of the surface layer is accounted for, but not in the unrelaxed surface. We used a (3, 3, 1) $k$-mesh to perform the structural relaxation and a (6, 6, 1) $k$-mesh for subsequent self-consistent field calculations. The (P)DOS plots are produced with the VASPKIT code[32]. Band structure unfolding onto the primitive cell was performed using the b4vasp package[33]. STM images were simulated using the Critic2 package[34]. Structural setup and visualization were performed using the Atomic Simulation Environment package[35].

## Data availability
Underpinning data will be made available online at ref. 36.

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

## Acknowledgements

We thank Sota Kitamura, Takashi Oka, Masafumi Udagawa and Hidetomo Usui for useful discussions and preliminary calculations. We gratefully acknowledge support from the UK Engineering and Physical Sciences Research Council (Grant Nos. EP/S005005/1 and EP/T02108X/1), the European Research Council (through the QUESTDO project, 714193), the Leverhulme Trust (Grant No. RL-2016-006), the Royal Society through the International Exchange grant IEC\R2\222041 and the Italian Ministry of Research through the PRIN-2022 "SORBET" project No.2022ZY8HJY. C.M.Y. acknowledges support from Shanghai Pujiang Talent Programme 21PJ1405400 and TDLI Start-up Fund. S.S. acknowledges the financial support provided by the Ministry of Education, Science, and Technological Development of the Republic of Serbia. S.S and S.P. acknowledge the CINECA award under the ISCRA initiative, for the availability of high-performance computing resources and support. We thank Diamond Light Source for access to the I05 beamline (Proposal No. SI28445), which contributed to the results presented here.

## Author contributions

C.M.Y. performed the STM experiments. G.R.S., P.A.E.M., T.A., and P.D.C.K. performed the ARPES experiments. S.S. and S.P. performed theoretical modelling. S.K. and A.P.M. provided samples. C.M.Y. analysed the STM data, and G.R.S. the ARPES data. I.B. performed preliminary calculations. M.D.W. maintained the Diamond I05 nano-ARPES beamline and provided experimental support. C.M.Y., G.R.S., S.S., S.P., P.D.C.K., and P.W. wrote the manuscript. All authors discussed the manuscript. The project was initiated by P.W., P.D.C.K., and A.P.M.

## Competing interests

The authors declare no competing interests.
