## [Peer Review File · Nature Communications]

Avoided metallicity in a hole-doped Mott insulator on a triangular lattice

Editorial Note: Parts of this peer review file have been redacted as indicated to maintain the confidentiality of unpublished data.REVIEWER COMMENTS

Reviewer #1 (Remarks to the Author):

The manuscript by Chi Ming Yim et al. presents the differentiation of the two distinct surface terminations in cleaved PdCrO₂ by using two separate surface-sensitive techniques, i.e. photoemission spectroscopy and STM, and reports a charge disproportionated insulator state in the CrO₂-terminated surface of PdCrO₂, which is taken as a hole-doped Mott insulator. The absence of metallicity by hole-doping a Mott insulator is unexpected and interesting in physics, calling upon a new understanding.

Yet, the conclusion---a reconstruction-induced charge disproportionated insulator, is likely of interest to a limited audience. Despite this, the key evidence is basically speculative based on calculations and the supports provided from the experimental side are unconvincingly connected, making the manuscript improper for publication in Nature Communications in the present form.

The authors probed an insulating gap by STS at the reconstructed, glassy CrO₂ terminations. The key to draw the claimed conclusion is that the CrO₂-terminated surface is hole-doped. This is indirectly probed by photoemission spectroscopy, by which regions with distinct hole doping levels are found and were cataloged as CrO₂- and Pd-terminated layers. However, the results can be also attributed to the electron and hole puddles accompanied with excess Pd impurities, due to the inhomogeneity of the sample crystallinity at a μm scale. This is particularly reasonable when further considering that the patches with distinct hole dopings show amazingly large sizes $\gg 50\mu\text{m}$, which is far beyond the size limit that a mechanically cleaved surface with a single termination can reach.

Besides these concerns, there are other issues to be clarified:

- 1) For the XPS-measured data, both Pb- and CrO₂-terminated surface regions show Cr 3p core level, meaning the sublayer signal is probable, but the Pd 4p peak is absent in CrO₂-terminated region. Why?
- 2) The comparison between Fig 2f and 2g appears not satisfying, especially given that the Cr shoulder feature is rather too weak and invisible in Fig 2f.
- 3) Fig 3b agrees with the STM-imaged reconstructed CrO₂ surface, but only in a qualitative way, if we assume Cr A is invisible by STM. Furthermore, a simulated STM image based on Fig 3b with larger spatial size is recommended for a direct comparison with experiment.

4) The difference in the bias-dependent STM topography normally suggests an electronic-, rather than a structural-type origin. The reconstructed CrO₂ surface collected at a small bias shows atomic resolution, but is different from the triangular lattice at normal setpoint, which is rather wired.

5) It remains unclear what mechanism is responsible for the scenario that the charge disproportionation and the accompanying structural modification make the surface CrO₂ layer maintain an insulating nature.

6) The data points in Fig 4h appear too scattered, and are few in quantity within the initial range, e.g. $0 < \Delta t < 3 \times 10^3$ s, making the exponential fitting to obtain the delay time inaccurate. Furthermore, line- or network-type marker in Fig 4a-d may be helpful to guide the comparison between these time dependent STM images.

Reviewer #2 (Remarks to the Author):

The manuscript presents STM, ARPES, and DFT+U results on PdCrO₂ to show that the CrO₂ surface layer in the CrO₂-terminated surface is surprisingly insulating despite of its hole-doped nature. It is accounted for the formation of a charge disproportionation in the CrO₂ surface layer with a reconstruction of $(\sqrt{7} \times \sqrt{7})R \pm 19.1^\circ$, which is supported by DFT+U calculations with relaxed atomic positions. This reconstructed ground state is quite susceptible to an applied voltage pulse in STM, thus showing glassy behavior.

Overall, the manuscript is well written and the authors' claims are quite persuasive. Furthermore, the observation of insulating behavior in a two-dimensional hole-doped Mott-Hubbard system may provide an exciting playground by doping the surface to produce many intriguing physical properties of strongly correlated systems. Thus, I strongly recommend publication in Nature Communications.

There are minor things to be corrected:

1. p.5: A reference for AgCrSe₂ is missing.
2. p.7: The gap edge positions in Fig. 2g may be about ± 0.25 eV not ± 0.35 eV.

Reviewer #3 (Remarks to the Author):

The manuscript by Yim et al. reports a joint experimental and theoretical effort to understand the surface physics of the delafossite material PdCrO₂. In the bulk, the CrO₂ layers are agreed to be Mott insulating, giving rise to a hidden Mott state within an overall metallic (from Pd layer) system.

The CrO₂ terminated surface, which should be connected to an effective hole-doped scenario due to a polar-catastrophe mechanism, remains insulating in experiment. The authors convey the message that this remaining insulating state with doping is based on the transition from a Mott insulator to a charge-ordered insulator upon surface doping.

While this is a challenging work on an interesting correlated material, some issues should be clarified:

- the Pd-terminated surface should account for 0.5 electron doping per Pd. Is this electron doping reflected in the surface k_F against the bulk? In other words does the straightforward metallic doping story when going from the bulk to the Pd-terminated surface hold?

- one weak point is given by the fact, that the charge-ordered state is not explicitly verified from experiment. Naively, would it not be possible to collect (even if small) tunneling current above the different Cr(A,B) sites? Are there any other ideas to get to this?

- in line with this, DFT+U of course, by design, likes symmetry-broken states, here of charge-order kind. In this regard, the authors are a bit short on the role of oxygen, could some charge differentiation be organized in a ligand-hole fashion?

Is magnetic order essential to reach the charge-ordered state numerically?

- is it possible to perform on top of the present surface terminations some additional impurity doping by placing adatom? This could perhaps further help to characterize the insulating layer.

- are there any other effects of the supposed charge-ordered state on the conducting Pd layer below?

- comment on DFT+U for bulk PdCrO₂: it seems that spin-polarization is necessary to open the hidden-Mott gap, which has also similarly been seen in previous spin-polarized DFT calculations (e.g. Scientific Reports 5, 12428 (2015)). Yet more advanced correlations treatment do not need magnetic order to open that gap and can account for a true metal-

transition with rising U.

In summary, this is interesting work which aims at revealing further details on the challenging delafossite material PdCrO₂. While there are some open questions concerning the match between theory and experiment, this text might still be suitable for publication in Nature Communications. The authors should

discuss the aforementioned points and try to improve their

understanding/message along these lines.

We would like to thank all reviewers for their time and constructive feedback on our manuscript. We provide below a point-by-point response to specific issues raised by the referees. With the submission, you should find versions of both the main manuscript and the supplementary with and without the changes marked up.

Reviewer #1 (Remarks to the Author):

The manuscript by Chi Ming Yim et al. presents the differentiation of the two distinct surface terminations in cleaved PdCrO₂ by using two separate surface-sensitive techniques, i.e. photoemission spectroscopy and STM, and reports a charge disproportionated insulator state in the CrO₂-terminated surface of PdCrO₂, which is taken as a hole-doped Mott insulator. The absence of metallicity by hole-doping a Mott insulator is unexpected and interesting in physics, calling upon a new understanding. Yet, the conclusion---a reconstruction-induced charge disproportionated insulator, is likely of interest to a limited audience. Despite this, the key evidence is basically speculative based on calculations and the supports provided from the experimental side are unconvincingly connected, making the manuscript improper for publication in Nature Communications in the present form.

We are pleased that the referee found our observation of the absence of metallicity in the heavily hole-doped surface layer here interesting and we thank the referee for their comments and suggestions on our paper. We stress that while our conclusion of charge disproportionation does come from a comparison of our experiments and calculations, the key finding of an insulating state in a heavily hole-doped system is an experimental fact. There are several clear-cut features in both, experiment and calculations, that drive us to the conclusion that this insulator forms due to charge disproportionation. First, we observe spectroscopically that we have an insulating surface layer, whereas natural expectations would suggest it should be metallic. Thus some form of correlated insulating state must occur. Our DFT calculations provide a natural explanation for this in terms of charge order, being the lowest energy state we have found computationally, and which naturally emerges when imposing precisely the reconstruction found experimentally in our STM measurements. We consider that this agreement of multiple features between two independent experimental probes and calculations point to more than just a speculative conclusion. We therefore feel that our paper provides strong evidence from a combined comprehensive experimental and theoretical standpoint of the formation of a charge-disproportionated insulator here.

The authors probed an insulating gap by STS at the reconstructed, glassy CrO₂ terminations. The key to draw the claimed conclusion is that the CrO₂-terminated surface is hole-doped. This is indirectly probed by photoemission spectroscopy, by which regions with distinct hole doping levels are found and were cataloged as CrO₂- and Pd-terminated layers. However, the results can be also attributed to the electron and hole puddles accompanied with excess Pd impurities, due to the inhomogeneity of the sample crystallinity at a μm scale.

This is particularly reasonable when further considering that the patches with distinct hole dopings show amazingly large sizes $\gg 50\mu\text{m}$, which is far beyond the size limit that a mechanically cleaved surface with a single termination can reach.

Our conclusion that we observe different surface terminations rather than just electron- and hole-puddles is based on our spatially resolved spectroscopic results both here, and from our extensive experience in working with the sister delafossite compounds and studying different surface terminations in these. The size limit of a single termination depends strongly on the chemistry of the material, so a general argument is impossible to make. For the delafossite oxides, we have extensively characterised the distinct terminations of the cleaved surfaces of the family of $(\text{Pd,Pt})(\text{Co,Cr})\text{O}_2$ compounds. In the Co-based systems, for example, extremely clear signatures of electron- and hole- doping of the CoO_2 and (Pd,Pt) surface terminations can be observed, with clear-cut spectral signatures that easily allow identifying any minority/mixed terminations [Nature, 549, 492 (2017), PNAS, 115, 12956 (2018)], and again consistent between ARPES and STM [Sci. Adv. 7, eabd7361 (2021), npj Quant. Mat. 7, 20 (2022)]. We show representative spatial mapping data from one such set of measurements in Figure R1 of this reply, where well-defined surface terminations are present with length scales $\gg 50 \mu\text{m}$, entirely consistent with the length scale of the [redacted] variations that we assign here for the CrO_2 - and Pd-terminated surfaces. With similar variations also in core-level spectra (see also below), we are confident in similarly assigning the observed patches here as from Pd- and CrO_2 -terminated regions, with minimal contribution from charge puddles due to excess Pd impurities. Note, however, that this does not mean that the surface terminations are atomically flat over this probing region. Indeed, during the STM measurement on the CrO_2 terminated surface we occasionally recorded large scale topographic images to check the spatial homogeneity of the cleaved surface. As shown

in Figure R2 of this reply, one such image with image size of $400 \times 400 \text{ nm}^2$ is characterised by flat terraces separated by step edges running along the same direction. The line-profile taken across the terraces shows clearly that the terraces are separated from each other by the same step height of $\sim 600 \text{ pm}$, indicating that all the terraces are of the same, i.e. CrO_2 termination. Importantly, however, there is negligible sign within this probing region of spatially varying Pd impurity contributions. We have added this figure as a new Supplementary Fig. 2 in the revised manuscript, and related discussion as supplementary note 2. Moreover, our STM data show no spatial inhomogeneity of the gap as would be expected if there were charge puddles. We also note that the ARPES measurements shown here were obtained using a probing light spot of $\sim 4 \mu\text{m}$, and so again from this and comparison between the STM and ARPES data, a significant impact of spatial inhomogeneity within the probing spot seems unlikely. We have worked to clarify this point in the revised text.

Figure R2: Large scale STM topographic image taken from the surface of a freshly cleaved PdCrO_2 single crystal sample ($V = 1.5 \text{ V}$, $I = 0.1 \text{ nA}$, Image size: $400 \times 400 \text{ nm}^2$). The image shows flat terraces separated from each other by step edges of the same step height.

More generally, regarding the level of anticipated doping, we can again make an informative comparison with the Co-based sister compounds. In our measurements shown in Figure R1, we observe the formation of distinct sets of surface states on the Pd and CoO_2 surface terminations. From a Luttinger analysis of the resulting Fermi surfaces, we find a doping of ~ 0.5 holes/Co atom on the CoO_2 -terminated surface and ~ 0.4 electrons/Pd atom on the Pd-terminated area. In the simplest ionic pictures, a doping of 0.5 holes or electrons would be expected, respectively. Deviations from this could occur due to additional interlayer (e.g. sub-surface) charge transfer and also from any inhomogeneous Pd adatom distributions of the form that the referee suggests. However, the lack of any significant deviations from the 0.5 e/h values observed in the structurally nearly identical Co-based compounds suggests that these effects are not significant for this materials class, and a nominal hole doping of the CrO_2 -terminated surface of close to 0.5 holes per Cr should be expected here. Finally, we note that the electronic structure observed on the distinct Pd- and Cr-terminated surfaces shown in Fig. 2 of the main text is not just exhibiting a small shift as might be expected from inhomogeneous Pd concentrations, but rather develops a substantially different structure with an approximately 500 meV shift in binding energy for the Cr states marked in Fig. 2f of the main text, and, importantly, also the development of a new low-energy shoulder. The fact that these distinct spectral signatures are closely matched to variations in the core

level data shown in Fig. 1 (cf. Fig. 2a-c) thus allows us to confidently assign them as signatures of the CrO₂-terminated surface. We have again worked to better highlight and clarify these points in the text.

Besides these concerns, there are other issues to be clarified:

1) For the XPS-measured data, both Pb- and CrO₂-terminated surface regions show Cr 3p core level, meaning the sublayer signal is probable, but the Pd 4p peak is absent in CrO₂-terminated region. Why?

As the referee points out, due to the finite probing depth in our photoemission experiment, signal from the subsurface layer can in general be expected. However, we also note that – at the probing photon energies used here – the photoemission cross-section for the Cr 3p core level is at least 2.5 times larger than for the Pd 4p core level. Thus the signal of the subsurface Pd 4p core levels can be [redacted] expected to be very weak, which is reflected in only a small shoulder feature being observed for this in the CrO₂-terminated regions. Indeed, we also observed similarly large differences in the intensity of the transition-metal and the Pd-derived core levels in our measurements from the PdCoO₂ sister compound (Figure R3 of this reply) where again only a weak subsurface Pd signal was obtained for the CoO₂-terminated surface. We also want to emphasize that the identification of the two surface terminations here is taken by the combination of a number of observable changes: the shift in the binding energy of the Cr 3p core level - in good agreement with the expected hole doping of this system; the spatial region where we observe a well-defined Pd 4p peak in the core level data, which anti-correlates with the Cr 3p core level shift; and the spectral signatures observed in our spatially-resolved ARPES measurements. We have elaborated on this further in the revised manuscript.

2) The comparison between Fig 2f and 2g appears not satisfying, especially given that the Cr shoulder feature is rather too weak and invisible in Fig 2f.

Figure R4: a) High resolution ARPES data of a CrO_2 - (left) and Pd-terminated (right) area of the sample. Both spectra have been probed using a photon energy of 110 eV and LH polarised light. The k -integrated spectra of the CrO_2 -terminated data is shown in b). By fitting the data a clear multi-peak structure can be revealed (see fit components 1-3, blue dashed lines) which is attributed to the presence of the bulk components of the Cr-derived states (labelled Cr) as well as the formation of a third surface component (S-Cr), which results in the observed shoulder in the k -integrated data. c) STS data (black dots) of the CrO_2 terminated surface of the sample taken within the sample bias range between -1.5 V and +1.5 V. (V_s, I_s): 1.5 V, 50 pA. ($f_{\text{mod}}, V_{\text{mod}}$): 413 Hz, 10 mV. Only data points in the occupied state region are shown. Black line: numerical fit to the STS data. Red dotted line: The fitted peak ($E = -0.43$ eV) associated with the Cr-derived surface states.

We agree with the referee that, at the aspect ratio of the plot in the main text, the shoulder is not immediately particularly pronounced in Fig. 2f. To help with identifying this, we plot in Figure R4 here the high-resolution data shown in the main manuscript with enhanced contrast for both surface terminations. Already in the raw data a clear development of the low-energy shoulder can be observed at a binding energy of approximately -0.5 eV (S-Cr in Figure R4a). These features are only present for the spectra measured from CrO_2 -terminated regions of the sample, and are completely absent in the data of the Pd-terminated surface. We show in Figure R4b the momentum-integrated data from the CrO_2 termination (reproducing Fig. 2f of the main text but shown here over a slightly extended energy range). The S-Cr shoulder component is evident as a change in the slope of the EDC in the vicinity of the Fermi level, and can be clearly resolved in peak fits to the data. Specifically, to describe the total valence band region in the fit (red solid line) requires (blue dashed lines) a pronounced S-Cr component as well as higher binding energy peaks. The so-determined binding energy of the S-Cr component is approximately -0.44 eV, which is in good agreement with the peak observed in our differential conductance spectra recorded with STM. In Figure R4c, we show from an independent fit of a tunnelling spectrum the peak related to the surface gap to be at -0.43 eV, practically identical to the energy of the S-Cr component found in photoemission. We have added Figure R4 and this associated discussion in the supplementary material of the manuscript to better clarify this to the reader (Suppl. Note 4 and Fig. S4).

3) Fig 3b agrees with the STM-imaged reconstructed CrO_2 surface, but only in a qualitative way, if we assume Cr A is invisible by STM. Furthermore, a simulated STM image based on Fig 3b with larger spatial size is recommended for a direct comparison with experiment.

We thank the referee for pointing out that the simulated STM images would further corroborate our DFT calculations. We are happy to follow their suggestion, and have simulated the STM images in the constant-current mode. By changing the bias voltage, we were able capture the two most prominent features of our experimental STM images: at larger bias voltages (200 mV) a perfect triangular lattice can be observed, with no evidence of $\sqrt{7} \times \sqrt{7}$ reconstruction (Figure R5a). On the other hand, at small bias voltages, when only the Pd-derived electronic states close to the Fermi energy and located inside the surface gap are probed (Figure R5b,c for 10 mV and 5 mV respectively), we clearly observe a difference in contrast due to the two types of Cr atoms, with a spatial variation that is consistent with the local contrast observed in our experimental measurements. We note that the contrast variation is more inhomogeneous in our experiments, however, consistent with the glassy nature of this state. Note that the STM images do not capture the Cr states directly (these states are gapped away) but instead the subsurface Pd states by tunnelling through the CrO_2 layer. We Figure R5 as Fig. S9 in the Supplementary Material and the corresponding discussion in supplementary note 7.

Figure R5: Simulated STM images. Simulated STM images obtained in constant-current mode at (a) $V=200\text{mV}$, showing a perfectly triangular lattice, and (b, c) at 10mV and 5mV , respectively, showing the $\sqrt{7} \times \sqrt{7}$ reconstruction.

4) The difference in the bias-dependent STM topography normally suggests an electronic-, rather than a structural-type origin. The reconstructed CrO_2 surface collected at a small bias shows atomic resolution, but is different from the triangular lattice at normal setpoint, which is rather wired.

We agree fully with the referee, and believe that this is one of the unique features of our results. Here, we can explain the disappearance of the contrast at more conventional setpoints by the constant excitation of structural changes: only at very small currents can we image the static structure, whereas for large currents and large bias voltages, the structure is fluctuating on time scales faster than what we can resolve. To demonstrate this point, here we show additional STM images obtained at different tunnelling conditions. Shown in Figure R6, a to c, the images taken in the sample bias range between 150 mV and 550 mV show perfect triangular lattice with no evidence of the order however some evidence of

fluctuations, e.g. in panel c where some streaky appearance can be seen. The image contrast becomes drastically different when the sample bias is decreased to values of only a few millivolts and small currents. As shown in Figure R6, **d** and **e**, the images taken at 10 mV or below exhibit the short-range ordered glassy structure instead of the perfect triangular structure. In addition, at roughly fixed sample bias (5-10 mV), as we further lowered the current by ~ 150 times (25 pA \rightarrow 0.15 pA), the streaks occasionally present in (**d**) become much less frequent in appearance in (**e**).

Figure R6: **STM topographic images of the CrO₂ terminated surface taken at different tunneling conditions.** a-c, Images taken at the same surface location at different sample biases and tunneling currents (V, I): (a) 550 mV, 250 pA, (b) 350 mV, 105 pA, (c) 150 mV, 50 pA. Image size: (8 nm)². The surface point defects showing up in the high sample bias images (a and b) become barely visible in the image taken at 150 mV (c). Images taken in this bias range (150-550 mV) show perfect triangular lattice of the CrO₂ surface. d-e, Images taken at energies in the vicinity of the Fermi level. Image size: (8 nm)². The image in (d) was recorded at (V, I)=10 mV, 25 pA, while that in (e) was taken with much lower tunneling current: (V, I)=5 mV, 150 fA. Note that the images in (d) and (e) were recorded using different frame angles. Both images show the glassy structure. However, due to the much lower current used, the image in (e) shows much less streaks.

All these images indicate that the perfect triangular lattice of the CrO₂ surface shows up when the surface is imaged at energies outside the surface gap and the glassy structure dominates when imaged at energies inside the surface energy gap. In addition, in order to image the glassy structure without any tip-induced disruption to the structure, one needs to perform imaging on the surface at very low sample bias (a few mVs) and extremely low tunnelling current (~ 200 fA). This at the same time indicates that the characteristic energy scale associated with any excitations taking place in the glassy structure is in the few milli-

electronvolt range. We have added the panels from Figure R6 and associated discussion in supplementary Fig. S5 and suppl. note 6.

5) It remains unclear what mechanism is responsible for the scenario that the charge disproportionation and the accompanying structural modification make the surface CrO₂ layer maintain an insulating nature.

As we argue in the text, we propose that the mechanism is due to localization of the charges driven by Coulomb repulsion, favouring the reconstructed state over the delocalized state. We have amended the text to bring this point out more clearly. Other mechanisms, such as Fermi surface nesting, would not result in a fully gapped Fermi surface and fully gapped density of states. In contrast, the fully insulating nature observed here strongly points to the correlated nature of the charge carriers and their tendency to avoid metallicity.

6) The data points in Fig 4h appear too scattered, and are few in quantity within the initial range, e.g. $0 < \Delta t < 3 \times 10^3$ s, making the exponential fitting to obtain the delay time inaccurate. Furthermore, line- or network-type marker in Fig 4a-d may be helpful to guide the comparison between these time dependent STM images.

We agree with the referee that it would be nice to have more data for shorter times. We note, though, that it is difficult to acquire data at those times because of the time it takes to acquire the topographic images themselves. This is rendered even more difficult by the extremely small currents at which these topographic images have to be acquired. Most importantly, what we demonstrate here is that we can probe the dynamics of the surface reconstruction, evidence of a glassy nature, but do not draw any further conclusions from the temporal behaviour. We hope that this will motivate future studies of the glassy dynamics. We would like to highlight that the key result of the paper is that the surface avoids becoming metallic despite being a doped Mott insulator.

Reviewer #2 (Remarks to the Author):

The manuscript presents STM, ARPES, and DFT+U results on PdCrO₂ to show that the CrO₂ surface layer in the CrO₂-terminated surface is surprisingly insulating despite of its hole-doped nature. It is accounted for the formation of a charge disproportionation in the CrO₂ surface layer with a reconstruction of $(\sqrt{7} \times \sqrt{7})R \pm 19.1^\circ$, which is supported by DFT+U calculations with relaxed atomic positions. This reconstructed ground state is quite susceptible to an applied voltage pulse in STM, thus showing glassy behavior.

Overall, the manuscript is well written and the authors' claims are quite persuasive. Furthermore, the observation of insulating behavior in a two-dimensional hole-doped Mott-Hubbard system may provide an exciting playground by doping the surface to produce many intriguing physical properties of strongly correlated systems. Thus, I strongly recommend publication in Nature Communications.

There are minor things to be corrected:

1. p.5: A reference for AgCrSe₂ is missing.

We thank the referee for pointing this out. We believe that the reference for AgCrSe₂ should be showing properly as ref. 19 in the current manuscript version.

2. p.7: The gap edge positions in Fig. 2g may be about ± 0.25 eV not ± 0.35 eV. *The precise energy depends on how one defines the energy of the gap edge, but we are happy to adopt the suggestion by the reviewer. We have updated this in the text.*

Reviewer #3 (Remarks to the Author):

The manuscript by Yim et al. reports a joint experimental and theoretical effort to understand the surface physics of the delafossite material PdCrO₂. In the bulk, the CrO₂ layers are agreed to be Mott insulating, giving rise to a hidden Mott state within an overall metallic (from Pd layer) system.

The CrO₂ terminated surface, which should be connected to an effective hole-doped scenario due to a polar-catastrophe mechanism, remains insulating in experiment. The authors convey the message that this remaining insulating state with doping is based on the transition from a Mott insulator to a charge-ordered insulator upon surface doping.

While this is a challenging work on an interesting correlated material, some issues should be clarified:

We thank the reviewer for their helpful comments for clarification, which we are pleased to address in detail below.

- the Pd-terminated surface should account for 0.5 electron doping per Pd. Is this electron doping reflected in the surface k_F against the bulk? In other words does the straightforward metallic doping story when going from the bulk to the Pd-terminated surface hold?

We thank the referee for raising this important point, but are somewhat limited in the conclusions we can draw for this surface from the current experiments. In previous studies of PdCrO₂, a distinct set of surface states has been observed for the Pd-terminated surface, which share a striking similarity with those observed on the Pd-terminated surface of the sister compound PdCoO₂ [PNAS, 115, 12956 (2018)]. In such a case, a Luttinger analysis revealed a good agreement with the expected doping of 0.5 electrons/Pd atom. However, in the present work, we do not observe such well-defined surface states for the Pd-terminated surface. We believe that this is due to the highly reactive nature of elemental Pd especially with H, which is a large part of the residual particles in a UHV chamber. We used micro-ARPES measurements here to isolate signatures from individual surface terminations. However, in the experimental setup which allows these measurements, we have previously observed that e.g. the Pd-terminated surface states of the sister compound PdCoO₂ age very quickly, becoming diffuse and ill-defined in the measurements. This makes an in-depth study of the surface states on the Pd-terminated surface experimentally challenging. As the focus of our study here lies on the CrO₂ terminated area of a sample, we initially performed time-consuming spatial mapping, and as a result do not clearly observe any surface states

of the Pd-terminated PdCrO₂ regions here. Thus, we consider that the Pd-terminated regions here most likely reflect the underlying bulk-like states. Similar measurements devoid of any surface states have also been reported previously on the sister compound PtCoO₂ [Science Advances 1, 1500692 (2015)]. The fact that we still see here small changes in k_F for the Pd-derived bulk-like states on the Pd vs. the CrO₂-terminated surface thus likely reflect underlying charge transfer with the *sub*-surface. Our calculations indicate a small charge transfer between the surface CrO₂ and the subsurface Pd layer (Supplementary Fig. S12) at the CrO₂-terminated surface. Such a charge transfer indicates an enhanced coupling between the Pd and Cr-derived states. This can be expected to lead to a reduced k_F (slight hole-doping on the subsurface Pd layer) of the Pd states measured at the CrO₂-terminated surface vs. the bulk Pd-derived states, as well as a reduced v_F (due to the increase in Cr character in these states). Both signatures are observed here. Figure R7a of this reply, we reproduce the spatial variations in the k_F of the Pd-derived bulk bands, illustrating a reduced k_F on a CrO₂ terminated surface as compared to the Pd-terminated areas. To allow for a direct comparison between the Pd-derived states probed on the two surface terminations we show the resulting peak positions from fits to the MDCs of the Pd-derived states shifted by their respective k_F (blue and yellow lines in Figure R7b). The resulting Fermi-velocity (v_F) of these states, fitted using a linear approximation, is plotted on top of the data (red and purple solid lines). For the CrO₂-terminated surface we find a $v_F=4.2\pm 0.1$ eVÅ which is slightly smaller than for the Pd-terminated surface where $v_F = 4.6\pm 0.1$ eVÅ. This is entirely consistent with a picture of finite sub-surface charge transfer from the CrO₂ terminating surface to the Pd layer beneath, and we have now included this figure and an expanded discussion on this point in the revised Supplementary Materials (Fig. S3/suppl. note 3).

Figure R7: (a) Spatial map of the Fermi velocity v_F obtained from our ARPES data (reproduced from Fig. 2c of the manuscript). (b) Dispersion extracted from the ARPES data at the positions of the diamonds in panel a. The Fermi velocity is obtained by fitting a straight line to the data. The graph shows a clear systematic difference between the Fermi velocity on the CrO₂ and Pd patches of the sample.

- one weak point is given by the fact, that the charge-ordered state is not explicitly verified from experiment. Naively, would it not be possible to collect (even if small) tunnelling current above the different Cr(A,B) sites? Are there any other ideas to get to this?

We would argue that the STM images shown in fig. 2h,i and fig. 4a do provide direct experimental evidence for the charge order, which we then sought to verify from calculations. With regards to spectroscopic confirmation of the partial density of states at the A- and B-sites, due to the susceptibility of the glassy structure to the tunnelling condition in STM/STS measurements, we are only able to perform tunnelling spectroscopy measurements on the CrO₂-terminated surface of PdCrO₂ at low sample bias, with the spectroscopy set-point set at low bias voltage and current values (20 mV, 100 pA), to minimize the possibility of disruption to the glassy structure in the measurement process. In Figure R8, we show a topography in panel a, with the positions of bright and dark atoms marked in panel b. Figure R8c shows spectra averaged over the dark and bright atoms in b with a light blue and red line, respectively. We find that the spectra of regions composed of 'bright' and 'dark' Cr atoms differ in the intensities of the peaks near the Fermi level. We have included the Figure R8 (as Fig. S6) and relevant text in the revised supplementary materials (suppl. note 5) to enhance the completeness of the manuscript. We note that recording spectra in a wider energy range, which would be required for a detailed comparison with the partial density of states from DFT calculations, is not straightforward, because the tunnelling electrons start to trigger changes in the order (compare fig. 4 of the main text).

Figure R8: Tunneling spectroscopy on the CrO₂ terminated surface. **a**, atomically resolved STM topographic image taken from the CrO₂-terminated surface of PdCrO₂ ($V = 20$ mV, $I = 30$ pA, Size = 4×1.5 nm²). **b**, As (**a**), with overlaid dots of different colours marking the positions of the 'bright' (red) and 'dark' (light blue) Cr ions at which point differential conductance spectra $g(V)$ were taken. **c**, Averaged $g(V)$ spectra taken from the 'bright' (red) and 'dark' (light blue) Cr atoms. Spectroscopy setpoint $V_s = 20$ mV, $I_s = 100$ pA. Frequency and amplitude of bias modulation used $f_m = 413$ Hz, $V_m = 1$ mV.

- in line with this, DFT+U of course, by design, likes symmetry-broken states, here of charge-order kind. In this regard, the authors are a bit short on the role of oxygen, could some charge differentiation be organized in a ligand-hole fashion?

The significant difference between the magnetic moments of CrA and CrB points to the primary role of transition metals in the charge-ordering. However, because the wave functions of the associated states are hybridized across chromium and several surface oxygen sites, it is hard to say whether there occurs also a related ligand order. We have added a sentence to the manuscript.

Is magnetic order essential to reach the charge-ordered state numerically?

The atoms in the charge-ordered states do acquire a modified magnetic moment, however the specific magnetic order does not matter – different magnetically ordered ground states are energetically almost degenerate, while the biggest energy gain for the ordered state is realized through the structural changes. Thus, we consider that the charge order, rather than magnetism, is the primary driver of the insulating ground state of the surface layer observed here.

- is it possible to perform on top of the present surface terminations some additional impurity doping by placing adatom? This could perhaps further help to characterize the insulating layer.

This is in principle possible and would be an interesting extension. However, it comes with significant experimental challenges, and so we consider this to be beyond the scope of our present work. We would hope that future experiments can address this and establish what the correlated state in the metallic phase realised by doping would be.

- are there any other effects of the supposed charge-ordered state on the conducting Pd layer below?

Our DFT calculations show that there is finite charge transfer between the CrO₂ surface layer and the Pd layer underneath (see also discussion above). However, the electronic states in the Pd layer form a highly itinerant system, and so we expect the charge disproportionation to be screened out at very short length scales. We also note that the hybridization with the chromium states is very small. From the DFT calculations, Pd remains non-magnetic and the charge on all Pd atoms in the second layer is the same. We have added a sentence clarifying this point in the manuscript.

- comment on DFT+U for bulk PdCrO₂: it seems that spin-polarization is necessary to open the hidden-Mott gap, which has also similarly been seen in previous spin-polarized DFT calculations (e.g. Scientific Reports 5, 12428 (2015)). Yet more advanced correlations treatment do not need magnetic order to open that gap and can account for a true metal-to-metal transition with rising U.

We agree with the referee, in [Phys. Rev. Mater. 2, 085004 (2018)] the author shows that the Mott gap in bulk PdCrO₂ opens without the need to invoke magnetic order. We have a sentence to point out that this might be similar in the surface layer in the revised manuscript. We have also added the reference suggested by the referee.

In summary, this is interesting work which aims at revealing further details on the challenging delafossite material PdCrO₂. While there are some open questions concerning the match between theory and experiment, this text might still be suitable for publication in Nature Communications. The authors should discuss the aforementioned points and try to improve their understanding/message along these lines.

REVIEWER COMMENTS

Reviewer #1 (Remarks to the Author):

I appreciate the efforts made by the authors to revise the manuscript for a better presentation to benefit the readers. Part of my concerns are well addressed, but, I have to admit, not all of them as detailed below. The disorder-induced insulating state has been well established, leaving the work probably of rather limited interest to the community. Despite of this, the work might be publishable in Nature Communications as long as the following issues can be proven to play a negligible role in the main conclusion.

One of the remaining concerns is especially regarding whether the two different ‘patches’ in the ARPES spectral-function map (Figs 2a-c) can be assigned as two different surface terminations of PdCrO₂. This is the most essential base for the conclusion of the manuscript. The issue is partly addressed by the authors, by basing on the comparison with a sister compound and the doping-level analysis. Yet, the newly provided discussions likely fail to eliminate the possibility of charge puddles to explain the two different patches, if we note the following points.

i) In Fig 2de, the measured ARPES dispersions from the two patches show no difference but a rigid shift in the Fermi level, indicating they are likely of the same surface termination with different doping levels. In contrast, in the ARPES data (Fig R1) for a sister compound, the similarly presented data show a clear difference in dispersion besides the Fermi level shift, which can be safely assigned as two different terminations.

ii) By referring to (Pd,Pt)(Co,Cr)O₂ compound, the author thought CrO₂-terminated surface would contribute ~ 0.5 h/Cr, and for the Pd-terminated area, ~ 0.4 e/Pd. As such, even if the CrO₂ plane becomes charge-neutral when alternating with Pd layer in real PdCrO₂ material, it also appears relatively hole-doped compared with Pd terminations as seen from the ARPES-measured dispersions. Instead, the resulting gap now is totally of band insulator type.

iii) The k-integrated spectra for the two patches both show a Cr shoulder at low-absolute-energy side (Fig 2f), despite that the authors insist its absence throughout. This again indicates the two patches are of the same termination.

Reviewer #2 (Remarks to the Author):

The manuscript is publishable as is.

Reviewer #3 (Remarks to the Author):

The authors have taken the criticism from the referees seriously and elaborated on improving the manuscript with the help of additional data and refined writing.

While the data is still not perfectly conclusive concerning the conclusions put forward by the authors, one still assess this work as an important one, with lots of new insights into the challenging delafossite physics, especially in view of competing electronic states/orders at the surface. With this novel iteration, I therefore support publication in Nature Communications.

We thank all reviewers for their time and constructive and positive feedback on the revised manuscript. We provide below a point-by-point response to specific issues raised by the referees, typeset in italics and blue. Enclosed with the submission, you should find versions of both the main manuscript and the supplementary with and without the changes marked up.

REVIEWER COMMENTS

Reviewer #1 (Remarks to the Author):

I appreciate the efforts made by the authors to revise the manuscript for a better presentation to benefit the readers. Part of my concerns are well addressed, but, I have to admit, not all of them as detailed below. The disorder-induced insulating state has been well established, leaving the work probably of rather limited interest to the community. Despite of this, the work might be publishable in Nature Communications as long as the following issues can be proven to play a negligible role in the main conclusion.

One of the remaining concerns is especially regarding whether the two different ‘patches’ in the ARPES spectral-function map (Figs 2a-c) can be assigned as two different surface terminations of PdCrO₂. This is the most essential base for the conclusion of the manuscript. The issue is partly addressed by the authors, by basing on the comparison with a sister compound and the doping-level analysis. Yet, the newly provided discussions likely fail to eliminate the possibility of charge puddles to explain the two different patches, if we note the following points.

i) In Fig 2de, the measured ARPES dispersions from the two patches show no difference but a rigid shift in the Fermi level, indicating they are likely of the same surface termination with different doping levels. In contrast, in the ARPES data (Fig R1) for a sister compound, the similarly presented data show a clear difference in dispersion besides the Fermi level shift, which can be safely assigned as two different terminations.

We must respectfully disagree with this classification. If this were just a rigid band shift, the bands would have shifted by ca. 500 meV (as determined from e.g. our core level shifts), which is a large amount and would lead to a change in k_F of ca. 0.1 \AA^{-1} , substantially larger than that observed experimentally. We also note that the highly dispersive band measured at the two surfaces are not just rigidly shifted copies of each other, but exhibit subtle changes in their velocity, as shown in Supplementary Fig. S4 of the revised submission. Similarly, for the CrO_2 -terminated surface, we find that the spectral weight distribution of the Cr-derived valence band as evident in, e.g., an MDC centered at $E-E_F = -0.3 \text{ eV}$ is markedly different from the one at 500 meV higher binding energy on the Pd-terminated surface (Fig. R1 of this reply). This again points

Figure R1: Comparison of the extracted momentum distribution curves (MDCs) from the two surface regions (labelled Pd- and CrO_2 in a) at different binding energies ($E-E_F$) as indicated by the solid lines in the spectra shown in a). Clear differences in the extracted MDCs of the two regions with a ΔE of 500 meV can be observed.

to a distinct electronic structure as measured at the two surfaces. The fact that the changes are less clear than for the sister Co-based material reflects the fact that the CrO_2 layer in the bulk of PdCrO_2 is Mott insulating, while in the CoO_2 case it is band insulating. As a result, hole doping into these states at the surface leads simply to the formation of new itinerant states in the CoO_2 -based material, while for the CrO_2 -based compound strong correlations drive the formation of the charge disproportionated insulator as shown here.

Moreover, our STM measurements show no intrinsic inhomogeneity of the CrO_2 terminated surfaces. In Fig. R2a, b we include the Laplacian of the topography shown in the original Supplementary Fig. S2, and have included the Laplacian in the revised Supplementary Fig. S3. There is no evidence for charge puddles in the image, either around the point defects or else. This is in strong contrast to what is seen in materials that do exhibit charge puddles. For comparison, just to give two examples, STM images are shown in Fig. 2c, d which exhibit typical nanometer-sized charge puddles around defects.

Figure R2: (a) topographic STM image over multiple terraces with the CrO_2 termination of PdCrO_2 . (b) negative Laplacian of the image in (a). While point defects can be seen, there is no evidence for extended charge puddles, which STM should pick up, as seen, for example, in Sr_2IrO_4 or cuprate superconductors. (c, d) examples of cases where charge puddles are seen in doped Mott insulators in (a) $\text{Ca}_{1.94}\text{Na}_{0.06}\text{CuO}_2\text{Cl}_2$ (Nat. Phys. 8, 534) and (b) $(\text{Sr}_{1-x}\text{La}_x)_2\text{IrO}_4$ at $x=2.2\%$ (Nat. Phys. 13, 21).

ii) By referring to $(\text{Pd,Pt})(\text{Co,Cr})\text{O}_2$ compound, the author thought CrO_2 -terminated surface would contribute ~ 0.5 h/Cr, and for the Pd-terminated area, ~ 0.4 e/Pd. As such, even if the CrO_2 plane becomes charge-neutral when alternating with Pd layer in real PdCrO_2 material, it also appears relatively hole-doped compared with Pd terminations as seen from the ARPES-measured dispersions. Instead, the resulting gap now is totally of band insulator type.

We do not follow this comment. In the bulk, Cr should be in a 3+ charge state within an ionic picture. This gives a Cr d^3 configuration, namely a half-filled Cr t_{2g} shell which is Mott insulating. A hole doping of ~ 0.5 h/Cr should give a $d^{2.5}$ average charge count at the surface, which must be metallic at the level of band theory. The fact that we find the surface remains insulating cannot therefore be explained by a gap of band insulator type, but must reflect a correlated insulator state.

iii) The k-integrated spectra for the two patches both show a Cr shoulder at low-absolute-energy side (Fig 2f), despite that the authors insist its absence throughout. This again indicates the two patches are of the same termination.

The data shown in Fig. 2b of the main text are the direct results of fitting the EDCs across our spatial map, where we include a component for the shoulder which the referee mentions, and allow it to change in amplitude. We plot the resulting amplitude in Fig. 2b, indicating that the amplitude of the surface component, S-Cr, is negligible on the patches which we assign as Pd-terminated, while a clear weight develops on the patches we describe as Cr-terminated. Of course, the valence band spectra are broad, with some residual spectral weight tailing out towards the Fermi level even for the Pd-terminated surface, but the clear spatial correlation shown in Fig. 2b allows us to directly associate the development of a pronounced shoulder feature with the CrO_2 -terminated surface. We have worked to better clarify this point in the text.

We have added the MDC comparison shown in Fig. R1 as a new Supplementary Figure S2 in the

*manuscript. Together with the above discussions, we feel that this further strengthens our conclusions regarding the distinct surfaces which are probed by ARPES. In addition, we stress that we see no evidence for charge puddles of the form the referee suggests within the spatial scale probed by STM. Our STM measurements show clean surface terminations (comp. Fig. R2), and we can uniquely identify the CrO₂-terminated surface from work function measurements. Our spectroscopy for this surface shows that it supports the same insulating state as is evident in the photoemission, and from which we get consistency with calculations only when considering the charge disproportionation at the surface. Thus we stress that we have two **independent** experiments which **both** directly point to an insulating CrO₂ surface layer, and both of these underpin the conclusions of our manuscript.*

Reviewer #2 (Remarks to the Author):

The manuscript is publishable as is.

Reviewer #3 (Remarks to the Author):

The authors have taken the criticism from the referees seriously and elaborated on improving the manuscript with the help of additional data and refined writing. While the data is still not perfectly conclusive concerning the conclusions put forward by the authors, one still assess this work as an important one, with lots of new insights into the challenging delafossite physics, especially in view of competing electronic states/orders at the surface. With this novel iteration, I therefore support publication in Nature Communications.

We thank referees #2 and #3 for their further consideration of our manuscript, and are pleased that they recommended publication of this version in Nature Communications.

REVIEWERS' COMMENTS

Reviewer #1 (Remarks to the Author):

The authors have addressed my concerns satisfactorily. I now recommend to publish it in Nature Communications.